# Out-of-hospital cardiac arrest and ambient air pollution: A dose-effect relationship and an association with OHCA incidence

**Francesca Romana Gentile**[1,2], **Roberto Primi**[1], **Enrico Baldi**[2,3], **Sara Compagnoni**[1,2], **Claudio Mare**[4], **Enrico Contri**[5], **Francesca Reali**[6], **Daniele Bussi**[7], **Fabio Facchin**[8], **Alessia Currao**[1], **Sara Bendotti**[1], **Simone Savastano**[1]*, **Lombardia CARe researchers**[¶]

1 Division of Cardiology, Fondazione IRCCS Policlinico San Matteo, Pavia, Italy, 2 Department of Molecular Medicine, Section of Cardiology, University of Pavia, Pavia, Italy, 3 Cardiac Intensive Care Unit, Arrhythmia and Electrophysiology and Experimental Cardiology, Fondazione IRCCS Policlinico San Matteo, Pavia, Italy, 4 Agenzia Regionale dell'Emergenza Urgenza (AREU) Lombardia, Milano, Italy, 5 AAT Pavia - Agenzia Regionale Emergenza Urgenza (AREU) c/o Fondazione IRCCS Policlinico San Matteo, Pavia, Italy, 6 AAT Lodi - Agenzia Regionale Emergenza Urgenza (AREU) c/o ASST di Lodi, Lodi, Italy, 7 AAT Cremona - Agenzia Regionale Emergenza Urgenza (AREU) c/o ASST di Cremona, Cremona, Italy, 8 AAT Mantova - Agenzia Regionale Emergenza Urgenza (AREU) c/o ASST di Mantua, Mantua, Italy

¶ Membership of the Lombardia CARe researchers is listed in the Acknowledgments.
* s.savastano@smatteo.pv.it

**Data Availability Statement:** All relevant data are within the manuscript and its S1 Dataset files.

**Funding:** We did not receive any specific funding for this work.

## Abstract

### Background

Pollution has been suggested as a precipitating factor for cardiovascular diseases. However, data about the link between air pollution and the risk of out-of-hospital cardiac arrest (OHCA) are limited and controversial.

### Methods

By collecting data both in the OHCA registry and in the database of the regional agency for environmental protection (ARPA) of the Lombardy region, all medical OHCAs and the mean daily concentration of pollutants including fine particulate matter ($PM_{10}$, $PM_{2.5}$), benzene ($C_6H_6$), carbon monoxide (CO), nitrogen dioxide ($NO_2$), sulphur dioxide ($SO_2$), and ozone ($O_3$) were considered from January 1st to December 31st, 2019 in the southern part of the Lombardy region (provinces of Pavia, Lodi, Cremona and Mantua; 7863 km2; about 1550000 inhabitants). Days were divided into high or low incidence of OHCA according to the median value. A Probit dose-response analysis and both uni- and multivariable logistic regression models were provided for each pollutant.

### Results

The concentrations of all the pollutants were significantly higher in days with high incidence of OHCA except for $O_3$, which showed a significant countertrend. After correcting for temperature, a significant dose-response relationship was demonstrated for all the pollutants examined. All the pollutants were also strongly associated with high incidence of OHCA in

**Competing interests:** The authors have declared that no competing interests exist.

multivariable analysis with correction for temperature, humidity, and day-to-day concentration changes.

## Conclusions

Our results clarify the link between pollutants and the acute risk of cardiac arrest suggesting the need of both improving the air quality and integrating pollution data in future models for the organization of emergency medical services.

## Introduction

Out-of-hospital cardiac arrest (OHCA) is a leading cause of death in industrialized countries. Both in Europe and the United States more than 350.000 OHCA occur annually [1, 2] In Italy, an average of 60000 OHCA occur per year, with an incidence of 1 per 1000 inhabitants. OHCA mostly occurs in patients with structural heart disease, in particular to those with coronary heart disease (CHD). Notably, in 15% of cases OHCA represents the initial clinical manifestation of CHD [3]. However, 10 to 15% of OHCAs occur in the absence of a structural heart disease, especially in young age. Despite efforts made in recent years to prevent and treat cardiac arrest, only 10% survive until discharge [4]. Due to the high mortality rate, the identification of factors that may precipitate cardiac arrest represents a major challenge and efforts in the field of primary prevention and risk assessment are needed. Over recent decades, air pollution has been established as a potential trigger for OHCA. The global mortality rate attributed to air pollutions is estimated at 8.8 million/year and the loss of life expectancy (LLE) from air pollution surpasses that of smoking, all forms of violence, HIV/AIDS, parasitic, vector-borne, and other infectious diseases [5]. Several studies have tried to estimate the health effects and cardiovascular morbidity of short and long-term exposure to various air pollutants highlighting how pollutants may contribute due to multiple mechanisms, such as endothelial dysfunction, vasoconstriction and systemic inflammation. However, the relationship between specific air pollutants and OHCA remains controversial.

The primary aim of this study is to examine the impact of short-term exposure to particulate and gaseous pollutants on the incidence of OHCA in a vast metropolitan and rural area that encompasses four provinces of the Po Valley in Northern Italy, one of the most polluted areas in Italy and Europe [6], due to its levels of industrialization and high population density. The secondary aim of this study is to search for and describe a dose-effect relationship, which could help predict OHCA incidence based on the concentration of pollutants in a specific area.

## Materials and methods

### Setting

This is a multicentre, observational, retrospective study based on prospectively collected data from our regional cardiac arrest registry (Lombardia Cardiac Arrest Registry: Lombardia CARe; clinical trial identifier on clinicaltrials.gov NCT03197142, ethical committee approval reference 20140028219 by the Ethical Committee of the Fondazione IRCCS Policlinico San Matteo in Pavia), started in 2015. Data are collected according to the 2014 Utstein style [7]. This registry represents the largest in Italy as it prospectively collects automated data on OHCAs occurring in five different provinces of Northern Italy: Pavia (since January 2015),

Lodi, Mantua, Cremona (since January 2019) and Varese (since January 2020), resulting in a total covered population of more than 2.4 million inhabitants. Registry data were collected and managed using REDCap (Research Electronic Data Capture) tools hosted at Fondazione IRCCS Policlinico San Matteo [8]. REDCap is a secure, web-based application designed to support data capture for research studies, providing 1) an intuitive interface for validated data entry; 2) audit trails for tracking data manipulation and export procedures; 3) automated export procedures for seamless data downloads to common statistical packages; and 4) procedures for importing data from external sources.

Pre-hospital information of each OHCA is automatically captured from the data warehouse of the Agenzia Regionale Emergenza Urgenza (AREU), our regional Emergency Medical Service (EMS), filed in our database the day after the event and then checked and validated by the local EMS personnel of each province. For each hospital of the territory, a team of doctors manage information regarding both the in-hospital stay and the short and long-term outcomes. Pre-hospital, in-hospital and follow-up information are combined in the Lombardia Care Registry and data quality is performed by a dedicated Study Management Team.

## Study population

The study population includes all the patients with diagnosed OHCA from the 1st of January 2019 to the 31st of December 2019 within the provinces of Pavia, Cremona, Lodi and Mantua in the Po Valley (Fig 1). These provinces cover a territory of 7.863 km2 with urban, suburban and rural areas and a population of over 1.5 million in 2019. The Po Valley is one of the most important industrial and agricultural areas in Italy and has a high population density. Moreover, due to the Alpine and Apennine chains surrounding the area, there is scarce ventilation so that the stagnation of pollutants in the air is quite common, especially in the fall and winter. For the present study, only OHCAs of presumed medical aetiology have been considered assuming that pollution has no effect on other causes of cardiac arrest.

## Air pollution and meteorological data

Particulate and gaseous pollutants including particles with aerodynamic diameter smaller than 10 and 2.5 μm ($PM_{10}$, $PM_{2.5}$), benzene ($C_6H_6$), carbon monoxide (CO), nitrogen dioxide ($NO_2$), sulphur dioxide ($SO_2$), and ozone ($O_3$) were available from the regional agency for environmental protection (ARPA), which monitors the air quality in all 20 Regions of Italy. A total of 29 monitoring stations were distributed in the study territory; specifically, six monitors were in the province of Cremona, seven in Lodi, seven in Mantua, and nine in Pavia (Fig 1). Daily concentration of both $PM_{10}$ and $PM_{2.5}$ were measured by 24 and 10 stations respectively; hourly concentrations were provided by 27 stations for $NO_2$, 7 for CO, 10 for Benzene, 16 for $O_3$ and 12 for $SO_2$. To avoid any potential confounding environmental factors, we also took into account meteorological data such as hourly temperature and hourly relative humidity during the study period, also provided by ARPA.

## Statistical analysis

Categorical variables were compared with the Chi-square test and presented as number and percentage. Continuous variables were compared either with the t-test and presented as mean ± standard deviation or compared with the Mann-Whitney test and presented as median and interquartile range (IQR) according to normal distribution tested with the D'Agostino-Pearson test.

We calculated the daily mean concentration for each pollutant for every monitoring station and then the mean value for the study territory was obtained. The change of the mean

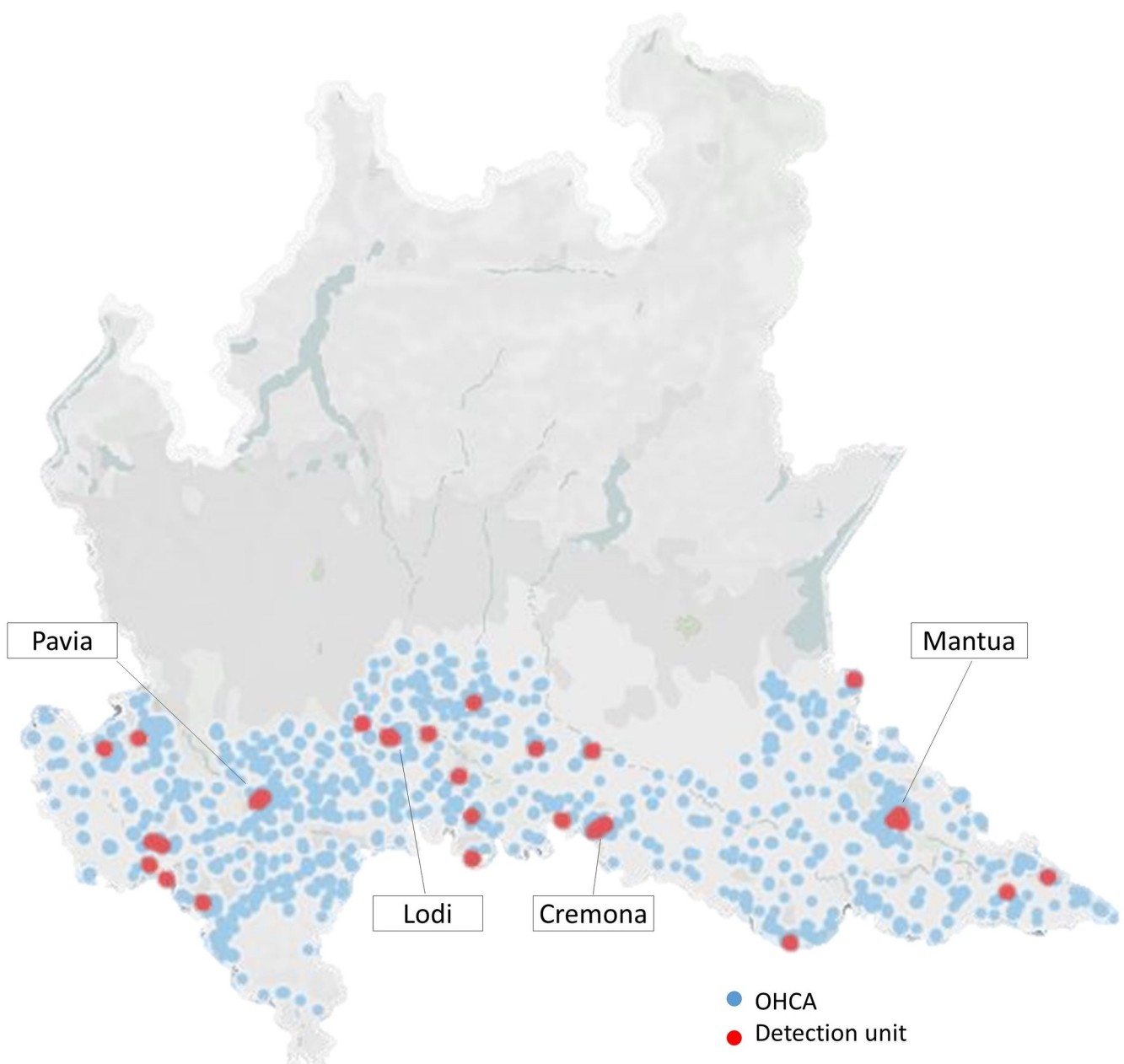

**Fig 1. Map of the Lombardy region.** The study territory in which OHCA are displayed as light-blue dots and monitoring stations as red dots. Created by Tableau Public software (Version 2020.3, LLC, Salesforce Company).

concentration between the index day and the previous day (day-to-day concentration change) was also calculated for every pollutant.

A correlation between the mean concentrations of every pollutant among the four provinces was tested via a multivariable regression model.

The daily incidence of OHCAs was calculated considering the daily number of OHCAs and the number of inhabitants in the study territory, monthly updated by the Italian institute of statistics (ISTAT), and then presented as cases per 100,000 inhabitants. The days of the year

were then divided into days with lower or higher incidence of OHCA according to the median value of the daily incidence.

An univariable logistic regression model was run for every pollutant to test the strength of association of the probability of having an incidence of OHCA higher than median value. We also ran a multivariable logistic model testing the strength of association of every single pollutant corrected for temperature, day-to-day concentration change and relative humidity.

A dose-response Probit regression model was run for each one of the pollutants, both before and after correction for temperature, where the dose was intended as the mean daily concentration of the pollutant and the presumed effect was the daily incidence of OHCA higher than the median value of the period.

Statistical analysis was performed with MedCalc software (Version 19.1.2 by MedCalc Software bv, Ostend, Belgium) and Fig 1 was created by using Tableau Public software (Version 2020.3, LLC, Salesforce Company). A p-value < 0.05 was considered statistically significant.

## Results

### OHCA incidence and population characteristics

Out of the 1922 EMS-assessed OHCA over the study period in the area of interest, 1582 had a presumptive medical aetiology (82%) according to the Utstein classification. The median daily incidence of OHCA in the overall territory was 0.3 cases/100.000 inhabitants.

The vast majority of OHCAs occurred at home. Less commonly, OHCA was reported to have occurred within nursing facilities, on the street, or in public places. The median age was 80 years old (IQR 68–87) and male sex accounted for 57% of cases. Resuscitation was attempted in 76.2% of the cases and ROSC was reached in 19.8%. The percentage of survived events was 19% but the survival to hospital discharge was achieved in 7.4% of the cases. Other characteristics of the study population are presented in S1 Table.

### Air pollution and incidence of OHCA

The trends of the atmospheric concentrations of the seven pollutants, temperature and humidity throughout the year were found to be highly statistically correlated among the four provinces of the study territory with multiple correlation coefficients (R-value) higher than 0.9. Only $SO_2$ showed an R-value of 0.4 which was still statistically significant (Fig 2).

Dividing the study period into days with high or low incidence of OHCA, according to the median value of the daily incidence (0.3 cases/100.000 inhabitants), we found that the concentrations of most pollutants were significantly higher in the days with a greater number of OHCAs as compared to those with a lower incidence. Benzene was the pollutant with the greatest difference [0.7 (IQR 0.4–1.2) vs 0.4 (IQR 0.3–0.7), p < 0.001], whereas $SO_2$ had the lowest and least significant difference between the two periods [3.2 (IQR 2.8–3.6) vs 3.1 (IQR 2.7–3.5), p = 0.046]. $O_3$ showed a countertrend, being significantly higher in the low-incidence period [29.9 (IQR 10.9–61.7) vs 56.1 (IQR 25.5–74.1), p<0.001]. Similarly, an increase in temperature was found significantly associated with the low-incidence period [10.1 (IQR 5.2–14.8) vs 15.1 (IQR 8.9–23.3), p<0.001] (Table 1).

In a univariable regression model, all but $SO_2$ were strongly associated with higher incidence of OHCA, while $O_3$ [OR 0.98 (95% CI 0.97–0.99) p<0.001], as well as temperature [OR 0.93 (95% CI 0.9–0.96) p<0.001], confirmed their countertrend (Table 2).

To avoid any confounding factors, we tested via a multivariable logistic regression model the strength of association of every pollutant after correction for the mean daily temperature, the mean daily relative humidity and for the day-to-day concentration change of the indexed

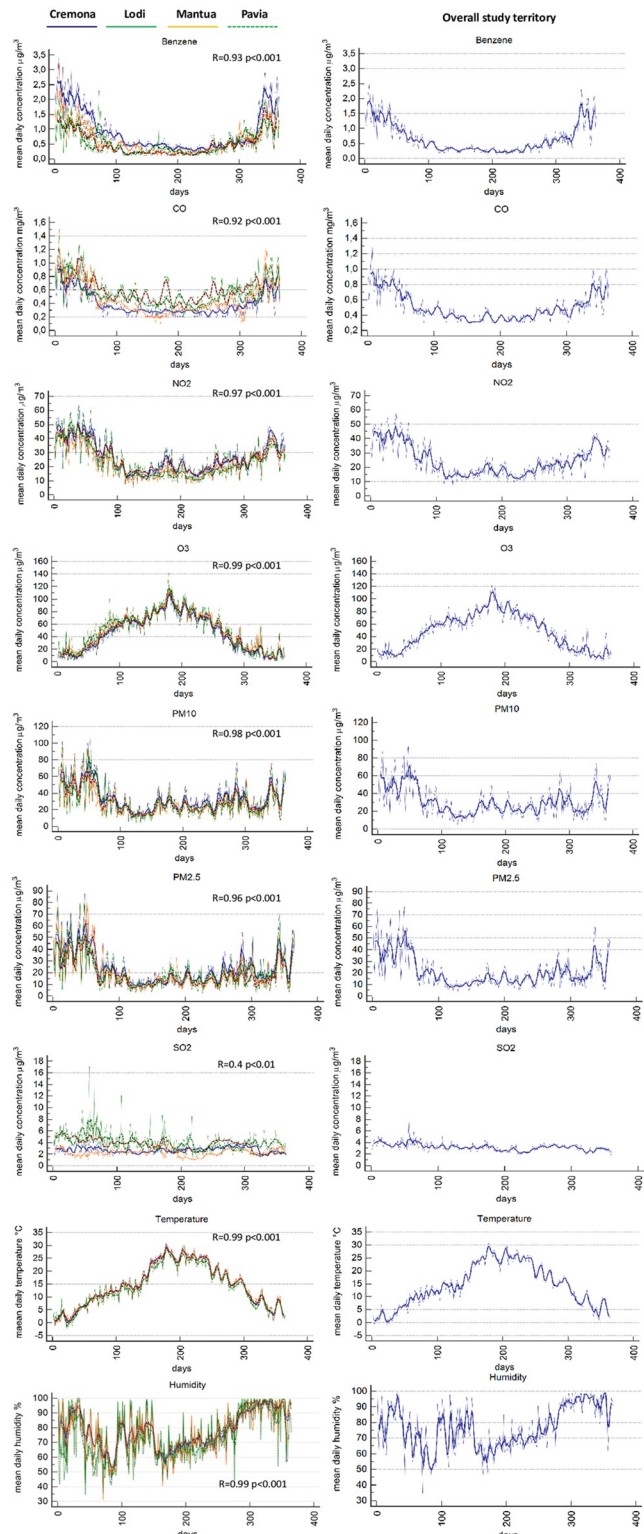

**Fig 2. Panels and trends.** The left panels depict the daily trend combined with the seven-days mean trend of each pollutant for every province. In the right panels the daily trend combined with the seven-days mean of all the study territory are displayed. The R value is the multiple correlation coefficient, resulting from the multiple regression analysis and referred to the correlation between the different provinces. R value is provided for every pollutant.

**Table 1. Air pollutants and meteorologic data.**

| Variable | Days with high incidence of OHCA (>0.3 cases/100000) | | Days with low incidence of OHCA (≤0.3 cases/100000) | | |
|---|---|---|---|---|---|
| | Median | IQR | Median | IQR | p value |
| Benzene (μg/m$^3$) | 0.7 | 0.4–1.2 | 0.4 | 0.3–0.7 | <0.0001 |
| CO (mg/m$^3$) | 0.5 | 0.4–0.7 | 0.4 | 0.4–0.5 | <0.0001 |
| NO2 (μg/m$^3$) | 27.4 | 18.4–39.6 | 20.2 | 15.1–27.6 | <0.0001 |
| O3 (μg/m$^3$) | 29.9 | 10.9–61.7 | 56.1 | 25.5–74.1 | <0.0001 |
| PM10 (μg/m$^3$) | 29.6 | 19.5–46 | 24.9 | 18.7–36.3 | 0.0219 |
| PM2.5 (μg/m$^3$) | 21.1 | 12.6–35.5 | 15.5 | 10.8–24 | 0.0040 |
| SO2 (μg/m$^3$) | 3.2 | 2.8–3.6 | 3.1 | 2.7–3.5 | 0.0464 |
| Temperature (˚C) | 10.1 | 5.2–14.8 | 15.1 | 8.9–23.3 | <0.0001 |
| Relative humidity (%) | 81.0 | 65.4–95.1 | 74.7 | 65.4–87.8 | 0.0778 |

Comparison of atmospheric mean concentration of each pollutant, temperature, and humidity between days with high incidence of OHCA (> 0.3cases/100000 inhabitants) and days with low OHCA incidence (≤ 0.3cases/100000 inhabitants).

pollutant. After correction, every pollutant tested demonstrated a statistically significant effect in being independently associated with higher incidence of OHCA. Interestingly, even $O_3$, which initially seemed to have an inverse relationship with OHCA incidence, was found to be strongly associated with high incidence of OHCA after these corrections. Moreover, $SO_2$, which in the previous analysis showed a non-significant effect, was shown to a be associated with higher OHCA incidence (Fig 3).

By using the Probit regression analysis, a dose-response relationship was demonstrated; before adjusting for temperature, the risk of OHCA increased significantly in relation to the elevation of the atmospheric concentration of all the pollutants analysed except for $SO_2$ (p = 0.08). $O_3$ confirmed its statistically significant negative relationship (p<0.001). However, based on the strong effect of temperature on OHCA incidence, the analysis was adjusted for temperature after which all the pollutants examined were found to be statistically significantly and directly related to a higher risk of OHCA. Specifically, $O_3$ inverted its initial trend (p<0.001) and $SO_2$ was found to be significant (p<0.001). (Fig 4).

**Table 2. Univariable logistic regression.**

| Univariable logistic regression model for the probability of having a higher incidence of OHCA (>0.3 cases/100000) | | | |
|---|---|---|---|
| Variable | OR | 95%CI | p |
| Benzene (μg/m$^3$) | 2.3 | 1.6–2.7 | <0.001 |
| CO (mg/m$^3$) | 10.6 | 3.3–36.8 | <0.001 |
| NO$_2$ (μg/m$^3$) | 1.04 | 1.03–1.07 | <0.001 |
| O$_3$ (μg/m$^3$) | 0.98 | 0.97–0.99 | <0.001 |
| PM$_{10}$ (μg/m$^3$) | 1.01 | 1–1.02 | 0.01 |
| PM$_{2.5}$ (μg/m$^3$) | 1.02 | 1–1.04 | 0.002 |
| SO$_2$ (μg/m$^3$) | 1.3 | 0.96–1.8 | 0.09 |
| Temperature (˚C) | 0.93 | 0.9–0.96 | <0.001 |
| Relative humidity (%) | 1.01 | 0.99–1.03 | 0.08 |

OR = odd ratio, CI = Confidence intervals, p = p-value.

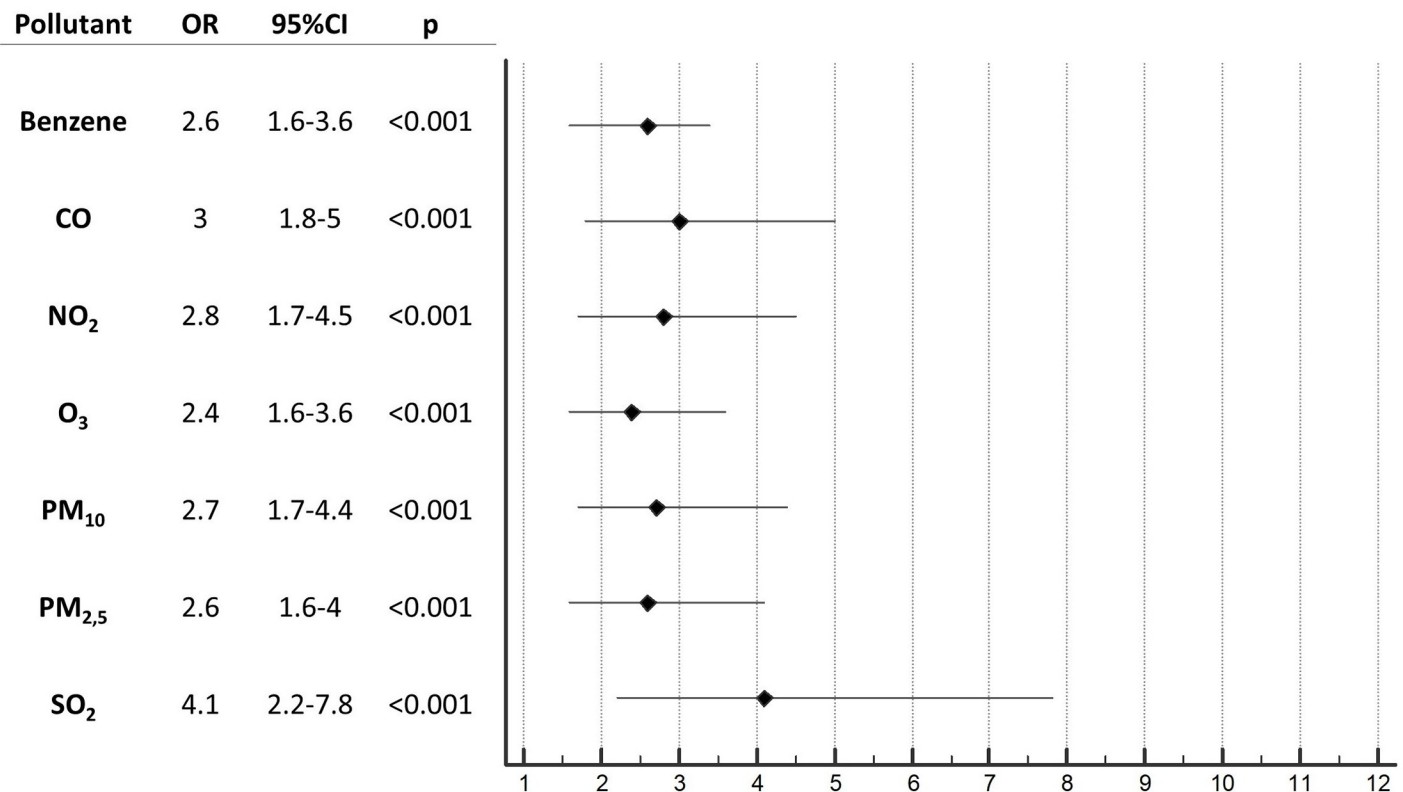

**Fig 3. Forest-plot.** Odds ratio resulting from multivariable logistic regression model for the probability of having a higher incidence of OHCA (>0.3 cases/100.000) after correction for temperature, day-to-day concentration change and humidity.

## Discussion

To the best of our knowledge, this is the first study to have provided a significant dose-response relationship between such a large number of air pollutants and an increased risk of OHCA. The identification of the shape of the exposure-curve is a key issue in decision making and strategic thinking in public health. All of the curves demonstrate that dose escalation leads to an increase in OHCA risk without showing any safe threshold (Fig 4). This is a reminder that large populations are exposed to differing levels of OHCA risk dependent upon the levels of pollutants in the specific area in which they live. To date, there are no threshold levels of pollutants advised as safe for the general population [9]. In our study, the concentration of every single pollutant examined correlated in the short-term throughout the year, even when the pollutants' concentrations were lower than the limit currently set by law in Italy or recommended by the World Health Organization's air quality guidelines (AQG). This statement has to be taken as a wish for a continuous slowdown in global air pollutant emissions. An exposure-response relationship has previously been described for PM2.5 and OHCA [10], but no dose-response curve has been ever found for all of the pollutants that we examined. Although the specific physiological effects of each of these pollutants is not fully understood at this time, cardiovascular health effects are gradually being established with controlled human and animal toxicological studies [9]. Similar mechanisms can be surmised to be responsible for respiratory disease caused by air pollution as well. Moreover, the health outcomes for CV and respiratory diseases associated with ambient air pollution are often overlapping in their aetiology and symptoms, and these two major groups of pathologies are often comorbid conditions. For this

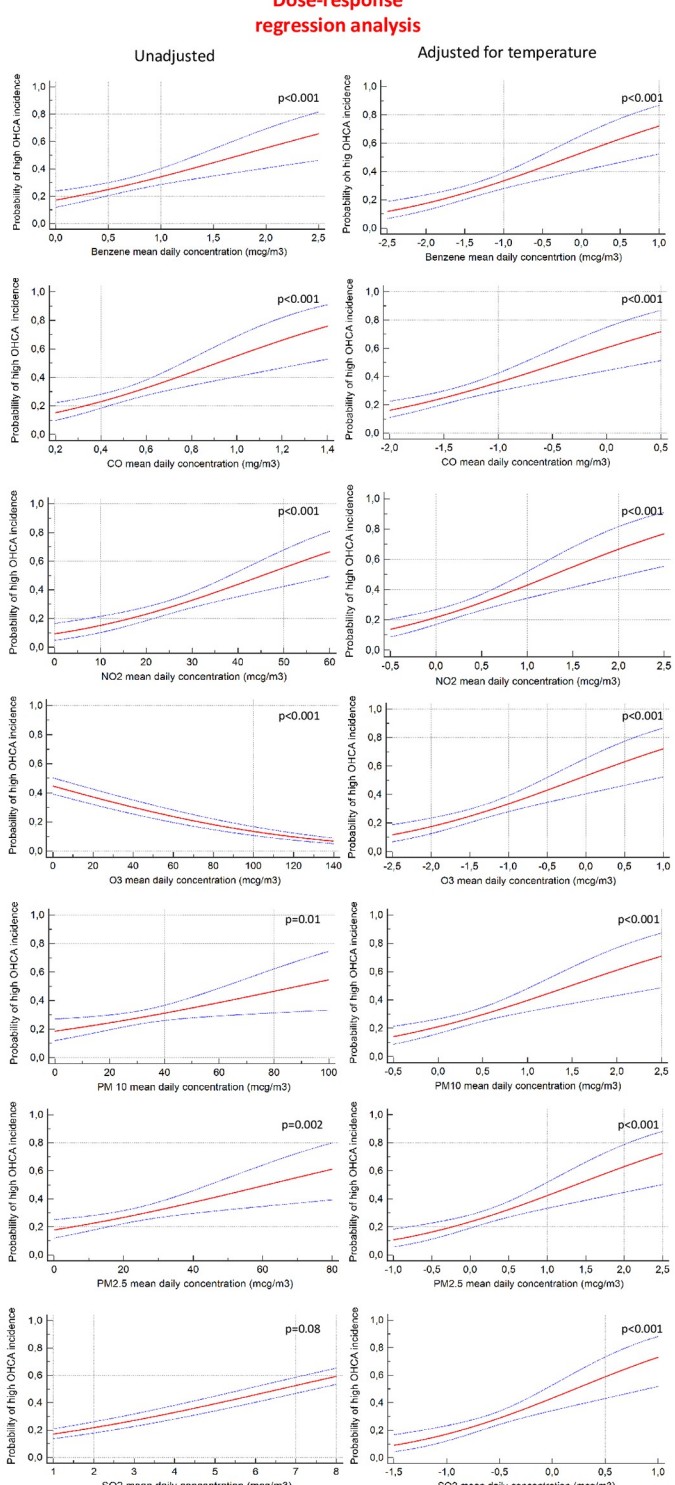

**Fig 4. Dose-response curves.** Probit analysis for every pollutant before (left panels) and after (right panels) correction for temperature.

reason, and for consistency with the 2014 Utstein style, we have decided to include all OHCA with presumptive medical aetiology, with no distinction between cardiac and respiratory causes, and to exclude cardiac arrests caused by external causes such as drowning, trauma, asphyxia, electrocution and drug overdose.

Importantly, this study has also demonstrated that the daily cumulative concentration of all seven pollutants influenced the percentage of OHCA occurring on the same day. This is in line with a recent meta-analysis, which assessed the effect of pollutant exposure with a distributed lag model [11] suggesting that the mean concentration of the current day and the previous day, which is alike our mean daily concentration from midnight to midnight, had the strongest estimated effect on OHCA risk of all of the lag patterns examined. This can be further interpreted to suggest that the cumulative short-term exposure to pollutants has a greater impact on OHCA risk than any lag pattern examined in previous studies. However, this meta-analysis confirmed a significant association only between short-term exposure to $PM_{2.5}$, $PM_{10}$ and $O_3$ and OHCA.

Several discrepancies can be found in literature regarding the effects of specific pollutants on CV health: acceptably robust data is present regarding particulate matter, while for gaseous pollutants, such as $O_3$, $NO_2$, Benzene, CO, and $SO_2$, the role in cardiovascular toxicity must still be better assessed. In a systematic review [11] based on 15 different studies conducted in North America, Europe, Australia and Asia, the association between the incidence of OHCA and $SO_2$ and CO were not found to be statistically significant. Also, in contrast to a large study conducted in Seoul [10], in which $O_3$ had no significant association with OHCA risk, a study conducted in Stockholm [12] demonstrated a significant association for $O_3$, but not for $PM_{2.5}$ and $NO_2$. By using a lagged exposure model, a recent study in Japan found that $PM_{2.5}$, CO, $O_x$ and SO2 were associated with higher risk of all-cause OHCA, while $NO_2$, was not [13] The reason why such a discrepancy might exist may be attributable to the differences existing in local climatic and socioeconomic conditions, demographic features and methodologies in which the studies were chosen to be conducted. Our significant results for all pollutants might be due to the high average concentrations of pollutants in the area of the study, one of the most polluted in Europe. The median value of $PM_{2.5}$ that we found in days with high incidence of OHCA was 21.1 µg/m3, in contrast to 8.7 µg/m3, 12 µg/m3, and 4.8 µg/m3 in Copenhagen, New York, and Melbourne, respectively. Another possible explanation for these differences in significance are differences in the prevalence of CVD, OHCA susceptibility and population age. Data from the WHO show that the mean European population age is amongst the highest in the world and Italy has one of the oldest populations in Europe. This means that we studied not only one of most polluted areas in the world but also one of the oldest, where pre-existing health conditions are also more likely to be frequent. Importantly, the correction for day-to-day concentration change, which we have applied in the multivariate logistic regression analysis, emphasizes that regardless of trends in pollution concentrations, it is the absolute dose on the current day which most affects OHCA risk and not the relative variation of concentration compared to the previous day. This also implies that the reduction of air pollution is expected to produce immediate improvements in health outcomes and not just for future generations, which has major potential implications on cost–benefit analyses.

We have confirmed that meteorological factors have to be taken into account when speaking about air pollution. As in other studies [14–16], we have demonstrated an inverse relationship between temperature and OHCA incidence by showing increases in wintertime. Conversely, relative humidity was found to have a neutral role, possibly due to its lower variability in the study territory compared to other areas where it has been studied, and found to be a risk factor for OHCA [17]. The correction of the analysis for the temperature was crucial. In univariable analysis and in the Probit regression dose-response model, $SO_2$ initially did not

appear to be associated with acute OHCA risk and $O_3$ showed a presumptive protective effect. More specifically, days with higher atmospheric concentrations of $O_3$ were associated with a lower incidence of OHCA, falsely suggesting that $O_3$ could have protective effects regarding OHCA risk. As shown in Fig 2, the majority of pollutants are present in increased concentrations during colder seasons because of home heating, increased use of transport vehicles and industrial activities. $O_3$ differs from this pattern and notably reaches higher atmospheric concentrations during the warmer months of the summer when the influence of direct sunlight is also the greatest. For this reason, we repeated our analysis after correction for temperature, humidity and day-to-day concentration changes and, after doing so, we unmasked the negative effect of $O_3$ and strengthened the correlation of all other pollutants, mainly of $SO_2$, which was found to have the strongest association [OR 4.1 (95% CI 2.2–7.8), p<0.001]. A study in Milan [18] has recently also demonstrated a synergistic effect between exposure to $PM_{10}$ and temperature. We believe that the interaction between air pollution and temperature deserves additional in-depth analysis, especially in a world in which climate change and global warming have become every-day topics affecting social behaviour and politics.

Currently, ambient air pollution is considered one of the greatest global health risks, causing significant loss of life expectancy, especially through cardiovascular disease, respiratory disease and cancer. It has been known for quite some time that the impact of inhaled pollution is more than just a pulmonary concern. At present, we know that some gases, such as NO and CO, are readily taken up into the circulation, while particulate matter, such as $PM_{2.5}$ and $PM_{10}$, are first filtered by the lungs [19]. In fact, acute air pollutant exposure has been demonstrated to increase the relative risk of hospitalization for heart failure, acute coronary syndromes, acute myocardial infarction (AMI) [20], cardiac arrhythmias, such as atrial fibrillation, and CVD mortality. A study conducted in Helsinki [21] suggested that air pollution triggers OHCA via two distinct modes: one associated with particulates leading to AMI and one associated with $O_3$ involving etiologies other than AMI, for example, arrhythmias or respiratory insufficiency. In the interim, numerous studies have tried to elucidate the physiological and molecular mechanisms involved, mostly focusing on oxidative stress, inflammation, and endothelial and autonomic dysfunction. For example, after acute $O_3$ exposure, alterations in heart variability and in QT duration, an increase of pro-inflammatory cytokines such as Interleukin-8 and Plasminogen activator-1 [22, 23], and an imbalance of autonomic components have all been noted [24]. Experiments have also found that acute $PM_{2.5}$ exposure can cause cardiac ischemia by crossing the alveolar-epithelial barrier into the circulation to affect cardiovascular function. In humans, controlled exposure of concentrated PM and to ozone led to brachial artery constriction, while CO has been noted to have the ability to induce right ventricular disfunction and pulmonary arterial hypertension [19]. Chronic exposure was similarly found to be related to an increase of individual cardio-metabolic risk due to epigenetic changes, including methylation, which affects transcriptional activity [25].

Since exposure to air pollutants is a modifiable factor that contributes to cardiovascular morbidity and mortality, the reduction of air pollution emissions is a valid strategy of primary prevention of CV disease and OHCA. However, due to the prevalence of other risk factors and existing differences in health care in other countries, it is difficult to assess the real magnitude of the benefits of such reduction.

Just as genetics, lifestyle and diet have a well-established role in CVD, environmental pollution is now playing an increasingly important role as well. In an era of precision medicine, air pollution exposure assessment may have a role in predicting OHCA susceptibility, especially for patients affected by other comorbidities [26]. The warning during high-polluted days should be addressed to these people in particular, even if to date there is no evidence about the role of previous hospitalizations in modifying the association between the risk of OHCA and

short-term increases in pollutant concentration [27]. By using a lagged exposure model, a recent study in Japan found that $PM_{2.5}$, CO, $O_x$ and SO2 were associated with higher risk of all-cause OHCA, while $NO_2$ was apparently found to be protective with OHCA incidence), and no differences in both sexes. [13]. In contrast to this, another study in Japan found that men were more susceptible than women [28] and in Singapore no clear evidence of increased susceptibility was found in elderly people as well as for people with cardiac history, diabetes mellitus or hypertension [20].

Ambulance service should also be planned accordingly on highly-polluted days since high pollution levels increases the demands of emergency services [29, 30]. The response speed of ambulance calls is very crucial to rescue patients having OHCA. Having more ambulances or EMS-professionals available in accordance with weather forecast and pollution monitoring could be useful in saving more lives. In a recent meta-analysis conducted in the UK [31], air pollution was significantly associated with an increase in ambulance dispatches, including those for cardiac arrest, asthma and other respiratory causes. A French study [32] examined the effects of short-term exposure to air pollution on the incidence of Mobile Intensive Care Unit (MICU) activation for cardiac arrest and it found a significant association between the increase of $PM_{2.5}$ and Ozone and the increased activation of MICUs for OHCA. Hence, air pollution monitoring could improve OHCA detection, response and care in transit. When high concentrations of pollutants are expected, more EMS resources should be available, which would lead to a more rapid dispatching of assistance and thus a more effective chain of actions leading to early cardiopulmonary resuscitation. Low-cost air pollution sensors may also be a future option to help reduce air pollution exposure and real-time measuring could be integrated in mathematical model for event prediction. These interventions could help in rendering the EMS model more efficient in order to optimize the prompt treatment of OHCA.

Some limitations should be addressed regarding this study. First, the 1-year time period in which this study was conducted may be insufficient to lead us to make general conclusions, especially towards finding a specific acceptable threshold level for the concentrations of individual pollutants. Second, the data used in this study comes from one of the most polluted areas of Europe, which might result in an overestimation of the influence of pollution on OHCA risk. Third, due to the lack of measurable biomarkers directly correlated to the individual exposure to the ambient pollutants, we have used air pollutant monitoring data, which might not represent actual exposure. Fourth, our register only documents where the OHCA occurred and patients' true exposure to pollutants may better correlate to other areas. However, the majority of the OHCA occurred at home and it is reasonable to think that those who have had OHCA in location other than home live in the same study area. Considering this, we believe that this aspect could only modestly affect our results. The current methodology also may not have captured the cumulative effects of the exposure or other effects derived from long-term exposure since it did not take lag effects into account. Pre-existing individual comorbidities were not accounted for, which might affect OHCA risk.

## Conclusions

A significant dose-response relationship was found for all seven of the pollutants studied and OHCAs, which provides important information to protect public health. All the pollutants have been demonstrated to have a strong association with OHCA incidence, independently from their day-to-day concentration changes and daily meteorological data. We truly wish for a time in which air pollutants monitoring can serve as a tool to better assess personal risk for OHCA and also to improve the general health service efficiency by being factored into ambulance forecast models and warning systems, both contributing to a reduction in OHCAs.

## Supporting information

**S1 Table. Summary characteristics for study population with out-of-hospital cardiac arrest.**
(DOCX)

**S1 Dataset. The study dataset of fully anonymized data.**
(XLSX)

## Acknowledgments

We would like to thank all the Lombardia CARe researchers, all the EMS personnel and a special thank you to Guido Giuseppe Lanzani and Orietta Cazzuli for ARPA Lombardia.

SS, EB and EC are members of the European Resuscitation Council Research-Net.

Lombardia CARe researchers: Antonio Cuzzoli, Andrea Pagliosa, Guido Matiz, Alessandra Russo, Andrea Lorenzo Vecchi, Cecilia Fantoni, Pierpaolo Parogni, Cristian Fava, Cinzia Franzosi, Claudio Vimercati, Dario Franchi, Enrico Storti, Ugo Rizzi, Simone Ruggeri, Erika Taravelli, Fulvio Giovenzana, Giovanni Buetto, Guido Francesco Villa, Marco Botteri, Salvatore Ivan Caico, Giuseppe Bergamini, Irene Raimondi Cominesi, Livio Carnevale, Matteo Caresani, Mario Luppi, Maurizio Migliori, Paola Centineo, Paola Genoni, Roberta Bertona, Roberto De Ponti, Stefano Buratti, Gian Battista Danzi, Arianna Marioni, Alessandra Palo, Antonella De Pirro, Simone Molinari, Vito Sgromo, Valeria Musella, Martina Paglino, Francesco Mojoli, Bruno Lusona, Michele Pagani, Moreno Curti, Catherine Klersy, Sabina Campi, Battistina Castiglioni, Umberto Piccolo, Marco Cazzaniga, Ilaria Passarelli, Giovanna Perone, Gianluca Panni, Daniele Ghiraldin, Luca Bettari, Luigi Moschini, Laura Zanotti.

## Author Contributions

**Conceptualization:** Francesca Romana Gentile, Simone Savastano.

**Data curation:** Roberto Primi, Enrico Baldi, Sara Compagnoni, Enrico Contri, Francesca Reali, Daniele Bussi, Fabio Facchin, Alessia Currao, Sara Bendotti.

**Formal analysis:** Francesca Romana Gentile, Simone Savastano.

**Investigation:** Roberto Primi, Enrico Baldi, Sara Compagnoni, Enrico Contri, Francesca Reali, Daniele Bussi, Fabio Facchin, Alessia Currao, Sara Bendotti.

**Methodology:** Simone Savastano.

**Software:** Roberto Primi, Enrico Baldi, Sara Compagnoni, Alessia Currao, Sara Bendotti.

**Supervision:** Claudio Mare, Simone Savastano.

**Validation:** Claudio Mare, Simone Savastano.

**Visualization:** Claudio Mare, Simone Savastano.

**Writing – original draft:** Francesca Romana Gentile, Simone Savastano.

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
