## [Decision Letter · Decision Letter 0]

2 Jun 2021

PONE-D-21-14468

Out-of-hospital cardiac arrest and ambient air pollution: a dose-effect relationship and a predictive role in OHCA risk.

PLOS ONE

Dear Dr. Savastano,

Thank you for submitting your manuscript to PLOS ONE. After careful consideration, we feel that it has merit but does not fully meet PLOS ONE’s publication criteria as it currently stands. Therefore, we invite you to submit a revised version of the manuscript that addresses the points raised during the review process.

Please address all the issues raised by the reviewers before re-submission.

We look forward to receiving your revised manuscript.

Kind regards,

Elena Cavarretta, M.D., Ph.D.

Academic Editor

PLOS ONE

Journal Requirements:

"Lombardia CARe is partially funded by the Fondazione Banca del Monte di Lombardia. Dr Baldi’s salary was partially

funded by grant 733381 from the European Union Horizon 2020 Research and Innovation Program of ESCAPE-NET.

4. One of the noted authors is a group or consortium [Lombardia CARe researchers]. In addition to naming the author group, please list the individual authors and affiliations within this group in the acknowledgments section of your manuscript. Please also indicate clearly a lead author for this group along with a contact email address.

5.We note that Figure(s)1 in your submission contain map images which may be copyrighted. All PLOS content is published under the Creative Commons Attribution License (CC BY 4.0), which means that the manuscript, images, and Supporting Information files will be freely available online, and any third party is permitted to access, download, copy, distribute, and use these materials in any way, even commercially, with proper attribution. For these reasons, we cannot publish previously copyrighted maps or satellite images created using proprietary data, such as Google software (Google Maps, Street View, and Earth). For more information, see our copyright guidelines: http://journals.plos.org/plosone/s/licenses-and-copyright.

a)    You may seek permission from the original copyright holder of Figure(s) 1 to publish the content specifically under the CC BY 4.0 license. 

Reviewers' comments:

Reviewer's Responses to Questions

**Comments to the Author**

1. Is the manuscript technically sound, and do the data support the conclusions?

Reviewer #1: Yes

Reviewer #2: Yes

2. Has the statistical analysis been performed appropriately and rigorously? 

Reviewer #1: Yes

Reviewer #2: Yes

3. Have the authors made all data underlying the findings in their manuscript fully available?

Reviewer #1: Yes

Reviewer #2: No

4. Is the manuscript presented in an intelligible fashion and written in standard English?

Reviewer #1: Yes

Reviewer #2: Yes

5. Review Comments to the Author

Reviewer #1: Authors have established statistically significant correlations between concentrations of air pollutants and out-of-hospital cardiac arrest. These correlations appear to follow a dose-response curve between concentration and number of events.

The quality of the writing is very good and the conclusions of the authors are sound. The discussion is well thought out. There are some easily modifiable points that can be addressed prior to publication:

1. The figures are of very low quality and should be replaced. Tables are fine.

2. In the regression analyses, presumably the concentrations published are the average over the course of 24 hours (but the y axis of these dose responses is not readable). Was there time-of-day data exposure available? Since cardiovascular events are somewhat rhythmically controlled, it would be interesting to see if these dose-responses were robust enough to correlate with hourly pollutant concentrations. If this is data is not available, perhaps only a brief mention of how the pollutant concentrations were reported.

3. Typo: (first paragraph of Air pollution and incidence of OHCA) ...was still statically significant (Figure 2)... - statistically significant

Reviewer #2: Introduction

- "look for a dose-effect curve" sounds very awkward, pls rephrase

Methods

- "A correlation between the mean concentrations of every pollutant among the four provinces was tested via a multivariable regression model.": Pls clarify how does this model work? How did you use a multivariable model to ascertain "correlation" between covariates?

- "We also ran a multivariable logistic model testing the predictive

power of every single pollutant corrected for temperature, daily concentration change and relative humidity": I think the logistic model when implemented in the manner you described, can inform on STRENGTH OF ASSOCIATION, not PREDICTIVE POWER.

- How did you account for seasonality? This is especially important for a 1-year study.

Results

- "R-values": I assume this is pearson's r correlation coefficient? It is confusing to state it as "R-value"

- Fig 2 & 4 are very blurry. I can't make out the words. I will need to examine again when clearer versions are provided.

Discussion

- limitations: the limitation of this simple log regression modelling approach, rather than standards in thie field eg DLNM need to be specified. for example, the inability to account for lag effects will lead to high risk of misclassification of dose-reponse. we know that the lag-specific risk for these pollutants are non-linear and often can be protective at some lag and harmful at some lag. this is a severe limitation of this study.

- Literature review: the discussion will be much improved by the inclusion of below literature

Ho, A. F. W. et al. Health impacts of the Southeast Asian haze problem – A time-stratified case crossover study of the relationship between ambient air pollution and sudden cardiac deaths in Singapore. International Journal of Cardiology 271, 352–358 (2018).

Kojima, S. et al. Association of Fine Particulate Matter Exposure With Bystander-Witnessed Out-of-Hospital Cardiac Arrest of Cardiac Origin in Japan. JAMA Netw Open 3, e203043–e203043 (2020).

Ho, A. F. W. et al. Time‐Stratified Case Crossover Study of the Association of Outdoor Ambient Air Pollution With the Risk of Acute Myocardial Infarction in the Context of Seasonal Exposure to the Southeast Asian Haze Problem. JAHA 8, (2019).

Zhao, B., Johnston, F. H., Salimi, F., Kurabayashi, M. & Negishi, K. Short-term exposure to ambient fine particulate matter and out-of-hospital cardiac arrest: a nationwide case-crossover study in Japan. The Lancet Planetary Health 4, e15–e23 (2020).

6. PLOS authors have the option to publish the peer review history of their article (what does this mean?). If published, this will include your full peer review and any attached files.

Reviewer #1: No

Reviewer #2: No

---

## [Author Response · Author response to Decision Letter 0]

13 Jul 2021

We thank the Editor and the Reviewers for their comments and their criticisms which helped us to improve the clarity and the quality of the manuscript.

Following is a point-by-point reply to the comments.

Journal Requirements:

A1: We fixed the style requirements including heading, citations, figures, tables based on the PLOS ONE style templates.

A2:We did not retract any papers. The only changes we have made to the reference list was made in consideration to the suggestion made by the Reviewer #2. In fact, we added 3 papers out of the 4 suggested. 

"Lombardia CARe is partially funded by the Fondazione Banca del Monte di Lombardia. Dr Baldi’s salary was partially

funded by grant 733381 from the European Union Horizon 2020 Research and Innovation Program of ESCAPE-NET.

A3:We are sorry for the inconvenience. We removed the funding-related text from the manuscript. The Funding statement should be the one stated in the Acknowledgments section. We will include it within the new cover letter.

4. One of the noted authors is a group or consortium [Lombardia CARe researchers]. In addition to naming the author group, please list the individual authors and affiliations within this group in the acknowledgments section of your manuscript. Please also indicate clearly a lead author for this group along with a contact email address.

A4: Thank you very much for your observation. At the time of the submission, we have probably misunderstood and made a mistake. In fact, we would like all the Lombardia CARe researchers to be listed in PubMed as done in a previous work (PLoS One. 2020 Oct 22;15(10):e0241028. doi: 10.1371/journal.pone.0241028. PMID: 33091034; PMCID: PMC7580972). The complete list of the researchers is in the acknowledgment section. 

5.We note that Figure(s)1 in your submission contain map images which may be copyrighted. 

All PLOS content is published under the Creative Commons Attribution License (CC BY 4.0), which means that the manuscript, images, and Supporting Information files will be freely available online, and any third party is permitted to access, download, copy, distribute, and use these materials in any way, even commercially, with proper attribution. 

For these reasons, we cannot publish previously copyrighted maps or satellite images created using proprietary data, such as Google software (Google Maps, Street View, and Earth). For more information, see our copyright guidelines: http://journals.plos.org/plosone/s/licenses-and-copyright.

a) You may seek permission from the original copyright holder of Figure(s) 1 to publish the content specifically under the CC BY 4.0 license. 

b) If you are unable to obtain permission from the original copyright holder to publish these figures under the CC BY 4.0 license or if the copyright holder’s requirements are incompatible with the CC BY 4.0 license, please either i) remove the figure or ii) 

supply a replacement figure that complies with the CC BY 4.0 license. Please check copyright information on all replacement figures and update the figure caption with source information. If applicable, please specify in the figure caption text when a figure is similar but not identical to the original image and is therefore for illustrative purposes only.

A5: As requested, the map has been created using “Tableau Public” which is a free platform that allows anyone to copy, distribute, and use every content created by their software. This uses map-sources such as OpenStreetMap® open data which is “licensed under the Open Data Commons Open Database License (ODbL) by the OpenStreetMap Foundation (OSMF)” and whose documentation “is licensed under the Creative Commons Attribution-ShareAlike 2.0 licence (CC BY-SA 2.0)”. https://opendatacommons.org/licenses/odbl/1-0/. As long as you credit OpenStreetMap and its contributors, everybody is free to “copy and redistribute the material in any medium or format” and to “remix, transform, and build upon the material for any purpose, even commercially” https://creativecommons.org/licenses/by-sa/2.0/. We added this information both in the method section and in the figure caption.

Reviewers' comments:

Reviewer's Responses to Questions

Comments to the Author

1. Is the manuscript technically sound, and do the data support the conclusions?

Reviewer #1: Yes

Reviewer #2: Yes

2. Has the statistical analysis been performed appropriately and rigorously?

Reviewer #1: Yes

Reviewer #2: Yes

3. Have the authors made all data underlying the findings in their manuscript fully available?

Reviewer #1: Yes

Reviewer #2: No

A6: You are right, the full database will be uploaded as a supplementary information.

4. Is the manuscript presented in an intelligible fashion and written in standard English?

Reviewer #1: Yes

Reviewer #2: Yes

5. Review Comments to the Author

Reviewer #1: Authors have established statistically significant correlations between concentrations of air pollutants and out-of-hospital cardiac arrest. These correlations appear to follow a dose-response curve between concentration and number of events.

The quality of the writing is very good and the conclusions of the authors are sound. The discussion is well thought out. There are some easily modifiable points that can be addressed prior to publication:

A1: Thank you very much for your kind appreciation. 

1. The figures are of very low quality and should be replaced. Tables are fine.

A2: Thank you very much for your comment. We have now improved the quality of the figures and replaced them as suggested.

2. In the regression analyses, presumably the concentrations published are the average over the course of 24 hours (but the y axis of these dose responses is not readable). Was there time-of-day data exposure available? Since cardiovascular events are somewhat rhythmically controlled, it would be interesting to see if these dose-responses were robust enough to correlate with hourly pollutant concentrations. If this is data is not available, perhaps only a brief mention of how the pollutant concentrations were reported.

A3: You are right in saying that in the regression analysis was entered the mean value of the day (over the course of 24 hours). As stated in the method sections: “daily concentration of both PM10 and PM2.5… hourly concentrations were provided…for NO2, …CO,…Benzene, …O3 and …SO2.” Concerning the idea of studying the hourly impact of the different pollutants, it was initially in our plans. However, we couldn’t perform this kind of analysis because not every station was able to provide the concentration of pollutants hourly. Infact, PM10 and PM2.5 were available only in daily concentrations.

3. Typo: (first paragraph of Air pollution and incidence of OHCA) ...was still statically significant (Figure 2)... - statistically significant

A4: Thank you very much, we fixed it.

Reviewer#2: Introduction

- "look for a dose-effect curve" sounds very awkward, pls rephrase

A1: As suggested, the sentence has been rephrased: “The secondary aim of this study is to search for and describe a dose-effect relationship, which could help predict OHCA incidence based on the concentration of pollutants in a specific area.”

Methods

- "A correlation between the mean concentrations of every pollutant among the four provinces was tested via a multivariable regression model.": Pls clarify how does this model work? How did you use a multivariable model to ascertain "correlation" between covariates?

A2: The reviewer is correct about this, and we have changed the term to “relationship” instead of correlation. We ran a multivariable regression model for each one of the pollutants and for both temperature and humidity, testing the relationship between the mean daily concentration of one province (dependent variable) with those of the other three (independent variables). What we found was a strong relationship and a good fit of the model with a “multiple correlation coefficient (R)” higher than 0.9 for all but one pollutant, and for temperature and humidity as well (Figure 2).

- "We also ran a multivariable logistic model testing the predictive

power of every single pollutant corrected for temperature, daily concentration change and relative humidity": I think the logistic model when implemented in the manner you described, can inform on STRENGTH OF ASSOCIATION, not PREDICTIVE POWER.

A3: We accepted the reviewer’ suggestion and we rephrased the sentence. According to this, we have also replaced the terms “predictor”, “predictive role” or similar with the more appropriate term “association”, both in the title of the study and along the entire text. 

- How did you account for seasonality? This is especially important for a 1-year study.

A4: We found that the temperature was indirectly associated with the incidence of OHCA and this probably reflects the seasonality of OHCA incidence. In fact, during summer the incidence of OHCA was lower than in the cold months of the year. However, during spring and summer the grade of pollution was lower as well. This is the reason that we have corrected the analysis for temperature. After this correction, every pollutant tested demonstrated a statistically significant effect on OHCA incidence. The most impressive effects of such a correction were concerning O3 and SO2: the former demonstrated a harmful effect on OHCA incidence only after correcting for temperature and the latter was only found to be statistically significant (vs non-significant) after correction.

Results

- "R-values": I assume this is pearson's r correlation coefficient? It is confusing to state it as "R-value"

A5: The R-value represents the “multiple correlation coefficient” of the multivariable model. We have added this information both in the fig 2 caption and the text.

- Fig 2 & 4 are very blurry. I can't make out the words. I will need to examine again when clearer versions are provided.

A6: We have now improved the quality of the figures and replaced them as suggested.

Discussion

- limitations: the limitation of this simple log regression modelling approach, rather than standards in thie field eg DLNM need to be specified. for example, the inability to account for lag effects will lead to high risk of misclassification of dose-reponse. we know that the lag-specific risk for these pollutants are non-linear and often can be protective at some lag and harmful at some lag. this is a severe limitation of this study.

A7: The reviewer is correct in saying that we did not take into account the lag effect. We acknowledge and agree that a distributed lag non-linear model is the most used way to evaluate the short and delayed effects of each air pollutant on OHCA incidence. However, in our study the difference of the concentrations of single pollutants between the index day and the previous one (formerly named “daily change in concentration” and now, for being clearer, named “day-to-day concentration change” did not show any major differences reflecting a constant high degree of environmental pollution in the study territory. We focused our attention on the immediate effect of the level of pollutants on OHCA incidence by considering daily average exposure/concentration on the day of the OHCA. This choice was done after having considered that the effects of many pollutants reached the maximum values with a lag pattern of 0-1, then decreased as the lag time increased. Also, as lag time increases the values tend to become less robust and more heterogenous. As you have rightly said, the percentage risk increase result in higher or lower values thus creating an over- or under- estimation of the effect on OHCA incidence. 

Aware of the limitation of our study model, we have also corrected the analysis for the variation of the concentrations between the index day and the day before. We are aware that our method can be limited by several factors, including the possibility to have not captured the cumulative effect of the exposure and the effects derived from long-term exposure. However, our results confirm those which others have described in the current literature and the dose-response curves were described. We have changed the limitations paragraph as suggested by the reviewer

- Literature review: the discussion will be much improved by the inclusion of below literature

Ho, A. F. W. et al. Health impacts of the Southeast Asian haze problem – A time-stratified case crossover study of the relationship between ambient air pollution and sudden cardiac deaths in Singapore. International Journal of Cardiology 271, 352–358 (2018).

Kojima, S. et al. Association of Fine Particulate Matter Exposure With Bystander-Witnessed Out-of-Hospital Cardiac Arrest of Cardiac Origin in Japan. JAMA Netw Open 3, e203043–e203043 (2020).

Ho, A. F. W. et al. Time‐Stratified Case Crossover Study of the Association of Outdoor Ambient Air Pollution With the Risk of Acute Myocardial Infarction in the Context of Seasonal Exposure to the Southeast Asian Haze Problem. JAHA 8, (2019).

Zhao, B., Johnston, F. H., Salimi, F., Kurabayashi, M. & Negishi, K. Short-term exposure to ambient fine particulate matter and out-of-hospital cardiac arrest: a nationwide case-crossover study in Japan. The Lancet Planetary Health 4, e15–e23 (2020).

A8:Thank you very much for your advice. We have made little changes along the discussion by adding the suggested papers to the references list.

---

## [Decision Letter · Decision Letter 1]

9 Aug 2021

Out-of-hospital cardiac arrest and ambient air pollution: a dose-effect relationship and an association with OHCA incidence.

PONE-D-21-14468R1

Dear Dr. Savastano,

We’re pleased to inform you that your manuscript has been judged scientifically suitable for publication and will be formally accepted for publication once it meets all outstanding technical requirements.

Kind regards,

Elena Cavarretta, M.D., Ph.D.

Academic Editor

PLOS ONE

Additional Editor Comments (optional):

Reviewers' comments:

Reviewer's Responses to Questions

**Comments to the Author**

1. If the authors have adequately addressed your comments raised in a previous round of review and you feel that this manuscript is now acceptable for publication, you may indicate that here to bypass the “Comments to the Author” section, enter your conflict of interest statement in the “Confidential to Editor” section, and submit your "Accept" recommendation.

Reviewer #1: All comments have been addressed

Reviewer #2: All comments have been addressed

2. Is the manuscript technically sound, and do the data support the conclusions?

Reviewer #1: Yes

Reviewer #2: Yes

3. Has the statistical analysis been performed appropriately and rigorously? 

Reviewer #1: Yes

Reviewer #2: Yes

4. Have the authors made all data underlying the findings in their manuscript fully available?

Reviewer #1: Yes

Reviewer #2: No

5. Is the manuscript presented in an intelligible fashion and written in standard English?

Reviewer #1: Yes

Reviewer #2: Yes

6. Review Comments to the Author

Reviewer #1: The authors have addressed my comments sufficiently and made a good attempt for the other, more rigorous reviewer as well

Reviewer #2: My concerns were satisfactorily addressed. The updated figures, which were of higher resolution, were satisfactory.

7. PLOS authors have the option to publish the peer review history of their article (what does this mean?). If published, this will include your full peer review and any attached files.

Reviewer #1: No

Reviewer #2: No

---

## [Editor Report · Acceptance letter]

16 Aug 2021

PONE-D-21-14468R1 

Out-of-hospital cardiac arrest and ambient air pollution: a dose-effect relationship and an association with OHCA incidence. 

Dear Dr. Savastano:

I'm pleased to inform you that your manuscript has been deemed suitable for publication in PLOS ONE. Congratulations! Your manuscript is now with our production department. 

Kind regards, 

on behalf of

Dr. Elena Cavarretta 

Academic Editor

PLOS ONE